# The Prospective Co-Parenting Relationship Scale (PCRS) for Sexual Minority and Heterosexual People: Preliminary Validation

**DOI:** 10.3390/ijerph19106345

**Published:** 2022-05-23

**Authors:** Daniela Leal, Jorge Gato, Susana Coimbra, Fiona Tasker, Samantha Tornello

**Affiliations:** 1Center for Psychology, Faculty of Psychology and Education Sciences, University of Porto, 4099-002 Porto, Portugal; jorgegato@fpce.up.pt (J.G.); susana@fpce.up.pt (S.C.); 2Birkbeck University of London, London WC1E 7HX, UK; f.tasker@bbk.ac.uk; 3Pennsylvania State University, State College, PA 16801, USA; slt35@psu.edu

**Keywords:** coparenting, sexual minority, prospective coparenting, stigma, familism

## Abstract

The coparenting relationship begins with a process of planning and negotiation about having children. Available psychological instruments have not been adapted to sexual minority people, which compromises their ecological validity. This mixed method study aimed to adapt and validate a prospective version of the Co-Parenting Relationship Scale in a Portuguese sample of sexual minority and heterosexual adults who did not have children and who were in a dyadic relationship. In study 1, cognitive interviews were used to gather participants’ reflections about the original items and the role played by the family of origin and anticipated stigma in coparenting (*n* = 6). In study 2, using a sample of individuals from 18 to 45 years old, two Exploratory Factor Analyses (EFA) were conducted separately for sexual minority (*n* = 167) and heterosexual persons (*n* = 198), and a Confirmatory Factor Analysis (CFA) was conducted for heterosexual persons (*n* = 176). Results showed underscored the importance of families of origin independent of sexual orientation. Different factorial structures for sexual minority and heterosexual persons were observed. Among sexual minority persons, the role of stigma was also highlighted. Implications for practice and research are discussed.

## 1. Introduction

Coparenting concerns the act of both members of the couple sharing family management, division of domestic labor, agreement on child education, and mutual support on parenting [1,2,3,4]. However, coparenting may also refer to divorced parents, persons who have children together but never were a couple, or two or more people who share the task of raising a child [5,6].

The quality of the coparenting relationship is associated with marital adjustment [7] and child outcomes [8]. Furthermore, supportive coparenting seems to enhance the couples’ relationship quality across the transition to parenthood [9]; it also represents an important mechanism to buffer some challenges in specific situations such as foster parenthood [10]. Disturbances in the coparenting relationship have been associated with development and psychopathology problems on children [5,11].

The coparenting relationship does not simply begin with the birth of children given that a process of planning and negotiation usually takes place before [5,6]. In fact, the way parents think about their future coparenting relationship seems to be a good predictor of their success in developing a strong coparenting alliance [5,6]. However, few studies have focused on the coparenting relationship across the transition to parenthood or in a prospective way [9,11].

Coparenting [12] among sexual minority people is also understudied [4], and most research has tackled the division of family labor among same-sex couples [13]. In fact, available instruments measuring the coparenting relationship have not been adapted to sexual minority people [14,15], which compromises their ecological validity [16]. As an exception, Carone and colleagues [17] analysed the psychometric properties of the Coparenting Scale-Revised [18] in a sample of Italian same-sex families. However, this instrument did not tap important coparenting dimensions such as the dyadic relationship or the division of labor [1,2,13,14,15] or specific challenges that sexual minority coparents may face, such as stigma [19,20,21,22,23].

The parenting experience, including the prospective one, which comprises the planning and negotiation about parenting and the way future parents think about their future coparenting relationship [5,6], also differs according to the social and cultural context [22,24,25,26]. Therefore, instruments assessing coparenting should encompass cultural variability [16,27]. In Portugal, where the present study was conducted, familistic cultural values prevail [28]. These values include cooperation, cohesion, and intergenerational support, which in turn, influence the journey into parenthood [22,24,25,29].

Thus, the aim of this study was to validate an existing instrument measuring the prospective coparenting relationship [2,12], with the added consideration of two previously unexplored dimensions of prospective coparenting: anticipated social support from the family of origin and, in the case of sexual minority individuals, anticipated social stigma. These additions will expand the concept of coparenting in a more inclusive way considering the specific challenges of LGB persons in coparenting. Furthermore, taking into account the importance of social support in this transition of life, support from families of origin is crucial to understand the prospective coparenting process.

### 1.1. Coparenting Relationship

Coparenting relationships have been conceptualized in different ways [2]. In Feinberg’s [1] framework of coparenting, five domains are distinguished: childrearing agreement, co–parental support/undermining, division of labor, joint management of family dynamics, and parenting-based closeness. Childrearing agreement refers to the similarity between parents’ view about how to raise a child. When there is a high level of disagreement about this topic, the opportunity for conflict will increase [1,2]. Co-parental support/undermining includes the acknowledgement and respect of the other parent’s contribution and promotion of their competency, decisions, and authority. The negative part of support is labeled as undermining and corresponds to criticism, disparagement, and blaming of the other parent [1,2]. Couples with positive relationships before the birth of a child seem to perceive the coparenting relationship as more supportive and less troubled [7]. Division of labor encompasses the sharing of responsibility, concerns, duties, and other childcare and household tasks.

Feinberg [1] proposed that the joint management of family dynamics could extend in at least three broad directions. First, parents might aggressively act out conflict and these interactions affect their parenting and, consequently, their children. Second, when the boundaries between the couple and their children are not clear enough, other family members could be co-opted into or excluded from parents’ relationship. Third, implicitly or explicitly there are expectations that define (i) how family members treat each other; (ii) the kind of structure and cohesiveness in family relations; (iii) the acceptance or avoidance parent–child coalitions; and (iv) the balance of parental interactions with children. Thus, parents differ in the degree to which they expected to be or are involved on whole-family interactions [1,2]. Lastly, the parenting-based closeness dimension includes the act of sharing the joy of parenthood and relates to the shared celebration of the child’s achievements of developmental goals, the experience of parents working together as a team, and regard for the partner’s development as a parent [2].

### 1.2. Coparenting, Social Support, and Stigma

Social support is linked to the quality of the coparenting relationship both directly and indirectly in several ways [1,15,30,31]. Extrafamilial social support and intergenerational relationships are expected to enhance parental adjustment independently of any connection to the coparenting relationship [30]. Specifically, Van Egeren [31] suggested that coparenting performance as modeled in families of origin will contribute to successful coparenting these reinforce positive experiences and contradict negative ones [32]. In the case of sexual minority parents, social support received from families of origin, friends, LGBT community, and other supportive networks during the transition to parenthood plays a crucial role for the mental health and well-being of lesbian, gay, and bisexual future parents [23]. Furthermore, sexual minority individuals who reported receiving more support from their families of origin also indicated stronger parenting alliances [15] and less parental stress [33]. Thus, it is important to understand coparenting taking into account the families’ ecological context and considering singularities that mark each co-parental relationship [34].

In addition, because families are embedded in social contexts, broader legal and cultural aspects also influence parenting. In the case of families headed by same-gender parents, the absence of legal recognition impacts both prospective [19,22,23,35] and current parenting processes [36,37,38]. As for the influence of culture, Leal et al. [22] found that, irrespective of sexual orientation, individuals without children in Portugal anticipated more social support in parenthood and less stigma if they decided to have children in comparison with counterparts in the UK. This seemed to apply to heterosexual and to LGB persons equally, with the more familistic culture of Portugal acting as a centripetal force pulling family members together across the generations [28,39]. Similar results were found by Shenkman et al. [26], when comparing the parenthood prospects in Portugal, the UK, and Israel. Thus, the social support from family of origin on coparenting plans should be inspected, and indeed may be an important factor in societies in which family social values are stronger.

### 1.3. Measurement of Coparenting

One of the most used multidimensional psychological instruments to assess the coparenting relationship is the Coparenting Relationship Scale (CRS) [2]. This instrument is composed of 35 items distributed by seven dimensions, namely: (i) coparenting agreement, (ii) coparenting closeness, (iii) exposure of child to conflict, (iv) coparenting support, (v) coparenting undermining, (vi) endorsement of partner’s parenting, (vii) and division of labor. This instrument presents good psychometric properties (reliability, stability, construct validity, and interrater agreement), and flexibly to be administered in short and long form [2]. In this regard, Pinto and colleagues [12] had adapt the Feinberg et al.’s [2] instrument in a Portuguese sample of prenatal fathers in a different-sex relationship.

### 1.4. The Present Study

In the present study, we aimed to adapt and validate a prospective version of the Co-Parenting Relationship Scale [2], specifically the Coparenting Relationship Scale—Prenatal Version (CRS-PV) [12] in a Portuguese sample of sexual minority and heterosexual adults in a dyadic relationship and who do not have children through a mixed methodology. We further hypothesized two additional dimensions of the prospective co-parenting experience: the social support from the family of origin and, in the case of sexual minority people, the specific challenges that coparents might anticipate related to stigma specific to entering parenthood as a partner in a same-gender couple relationship.

Three scales assessing the perception of coparenting relationship were already adapted to Portuguese language: authored by Carvalho et al. [40], Lamela et al. [41], and Pinto et al. [12]. The first two scales were not suitable for assessing coparenting prospectively and were thus discarded. The version by Pinto and colleagues [12] to pre-natal fathers was developed through sampling the reports of fathers who were expecting their first child within a maximum of nine months of gestation. We developed a new version of Pinto et al.’s [12] instrument to assure: (i) the adequacy to both sexual minority and heterosexual people; (ii) the prospective perception of coparenting; (iii) the role of family support and anticipated stigma upon parenthood. To guarantee the achievement of all proposed goals, we conducted the preliminary validation of Coparenting Relationship Scale—Prenatal Version (CRS-PV) [12] in two different studies. In Study 1, we used a qualitative methodology with semi-structured cognitive interviews to establish the facial validity of the instrument, and to develop items measuring social support and stigma upon parenthood. This process was conducted to assure the ecological validity of the instrument [16,27]. In Study 2, we conducted an Exploratory Factor Analysis (EFA) among a group of heterosexual participants to inspect the underlying relationships between the measured variables [42] This structure was further tested with two additional Confirmatory Factorial Analyses (CFA): one among sexual minority participants, and another among heterosexual participants (Figure 1).

## 2. Materials and Methods (Study 1)

### 2.1. Participants

Our convenience sample was recruited from the personal network of the first author and through the snowball technique. To be congruent with the criteria used to answer the instrument, participants must be in a committed relationship without children. To assure face validity, we recruited participants diverse in sexual orientation, gender, educational level, and age. Participants were asked to inspect the adequacy of the items of the CRS-PV [12] with regard to their own perceptions of coparenting, considering their sexual orientation, gender, and personal characteristics (ITC, 2018). Our sample was composed of six cisgender individuals, with half identifying as heterosexual and half as sexual minority people. All demographic characteristics are presented on Table 1.

### 2.2. Measures

Sociodemographic Questionnaire. Participants were asked about their age, ethnicity, gender identity, gender expression, sexual identity, educational attainment, relationship status, type of geographic location, and employment status. Completion of this questionnaire took no longer than five minutes.

The Coparenting Relationship Scale—Prenatal Version (CRS-PV) [12]. This scale is a Portuguese adaptation of the self-report measure constructed by Feinberg and colleagues [2]. It comprises 30 items distributed by four subscales, namely: (i) lack of coparenting support, (ii) coparenting conflict, (iii) coparenting disagreement, and (iv) coparenting undermining. The 30 items are scored on a seven-point Likert-type scale 1 to 7 (1 = Not true of us to 7 = Very true of us). It was adapted to assess coparenting during pregnancy and to our best knowledge previously has been used only with cisgender men during the transition to parenthood [12]. This instrument has different dimensions compared to the Coparenting Relationship Scale (CRS) [2]. First, the lack of coparenting support subscale assesses the individual perception of how the other parent gender specified by Pinto et al. [12] would be providing coparenting support (e.g., ‘I believe my partner will be a good mom’). Second, the coparenting conflict subscale assesses the perception regarding the likelihood of exposing children to parental conflicts (e.g., ‘Sometimes, one or both of us will say cruel or hurtful things to each other in front of the child’). Third, the coparenting disagreement subscale comprises the degree of disagreement expected regarding the child’s parenting (e.g., ‘My partner and I will have the same goals for our child’ (item reversed)). Lastly, the coparenting undermining subscale assesses the anticipation of critics, guilty and competition between the coparents (e.g., ‘My partner will undermine my parenting’).

Cognitive Interviewing [43]. The adequacy of the Coparenting Relationship Scale—Prenatal Version [12] was assessed using the following questions for which verbal answers were given: (i) This scale aims to assess the prospective coparenting relationship scale. Do you think this aim is achieved?; (ii) Were instructions of the scale clear?; (iii) Was length of the scale appropriate?; (iv) Did all items seem accurate?; (v) Did any item seem redundant?; (vi) Would you suggest the exclusion of any item?; (vii) Would you suggest the addition of any item?; (viii) Would you suggest the reformulation of any item?; (ix) Is there any other question you think that is important to be included in this scale?. Then, considering the reviewed literature about the importance of social support from families of origin, participants were asked: “When planning to be a parent, there are a lot of things we think about. For instance, the support from our own or our partners’ family might be important. Considering this, to what extent do you think the support from the family of origin is important for the coparenting relationship?”

Sexual minority people had a specific section of questions to be answered. First, they were asked if any questions raised difficulty for them in conjunction with their sexual orientation. Then, they were instructed “When we plan parenthood there are some challenges and questions that we anticipate and think about before achieving our parenting plans. Specifically, considering your sexual identity, what questions and challenges do you anticipate that might influence your coparenting relationship and that would not exist if you identified as heterosexual?”.

### 2.3. Procedure of Data Collection

This study was approved by the Ethic Committee of the host institution. Participants were asked to participate in an individual interview; three participants asked to be interviewed in a locality of their choosing and three were interviewed on the premises of the host institution. At the beginning of each interview, the researcher provided complete information about the study regarding all ethical issues, namely confidentiality, anonymity, the possibility to withdraw at any moment, and discussed how the data would be used. Subsequently, a written consent was read and signed by the participant, who also authorized the audio recording of the interview. There was no financial compensation for participation.

At the beginning of the interview, all participants were asked to answer the sociodemographic questionnaire and then the CRS-PV [12]. Then, considering the suggested guidelines of ITC [27] participants were invited to participate in a cognitive interviewing using retrospective probing technique [44] regarding the CRS-PV. Interviews lasted between 30 min and 45 min.

### 2.4. Procedure of Data Analysis

Interviews were first transcribed verbatim and checked for accuracy. To understand the participants’ reports, we decided to allow themes to emerge from the data. This procedure ensured the facial validity of the data [16,27,45]. Aside from the participants’ suggestions concerning particular items (cognitive interviewing), the emergent questions also generated with qualitative data was analyzed using thematic analysis [46]. This method allowed to identify and report patterns and themes within textual data.

Followed Braun and Clarke’s [46] six-step process for conducting thematic analysis: (i) familiarizing oneself with the data; (ii) generating initial codes; (iii) searching for the themes; (iv) reviewing the themes; (v) defining and naming the themes; (vi) producing the report. In the first step, the first author became familiar with the data by reading each transcript twice. Then, initial ideas for coding were raised, generating the initial codes in a second step. Step three involved putting the codes into potential themes, and the next step consisted of reviewing and refining the devised set of initial themes by checking whether the data fitted in each theme. In step five, all themes were clearly rendered, and the specifics of each theme were decided upon. In the final step, the report was written, and compelling excerpts from participants were selected to illustrate each theme. At this point, authors discussed and reached agreement on each theme.

## 3. Results

### 3.1. Cognitive Interviews

All participants suggested changes throughout the instrument, but no participants suggested the removal of any item. Globally, participants considered that many items were stated in a negative way and that this made their interpretation difficult. Furthermore, participants found it difficult to prospect some aspects of the coparenting relationship (i.e., to consider what their partner’s behaviors might be in specific situations). Several suggestions were made on item content and changes were made throughout the instrument. First, one participant suggested an introductory item to inspect to what extent the participants desired to have children. However, this item was not added as this question was already part of the larger research protocol. Second, because the instrument raises different questions about conflicts, one participant suggested the introduction of daily scenarios to facilitate the items’ interpretation. In this regard, another participant considered that the couples’ communication about problems might be an important dimension to be included in this instrument. Furthermore, two participants considered that the first item of the scale (“I believe my partner will be a good parent”), should be changed to other position given its confrontational content; then, participants suggested that item 25 should be presented first (“Parenting will give us a focus for the future”). Thus, item 1 and item 25 were switched. Item 3 raised some doubts among the participants because of its length; authors rearranged this item as follows: “My partner will ask my opinion about parenting issues”. The language used on item 5 was also considered confusing, and the item was changed to “My partner will like to play with our child and then he/she will leave the dirty work to me.” Concerning item 18, one participant, who had the lowest educational level, did not understand how to “compete for the child’s attention” could be considered a negative aspect. So, authors further emphasized the negative content of the item and changed it to “When all three of us will be together, my partner sometimes will want all our child’s attention for him/herself.” Regarding item 26, three participants considered that jokes and sarcasm might be a positive personal strategy to deal with challenges. Thus, this item was changed to “Sometimes, I will find myself in a mildly tense or sarcastic interchange with my partner”. Item 28 used the term “discuss” which raised some questions among four participants who had considered that this was not necessarily negative. Considering this, the item was changed to “We will argue about our relationship or marital issues unrelated to our child, in the child’s presence”. Three participants considered that items 26 to 30 shared a negative approach that may influence the interpretation of participants once the items were presented together. To overcome this difficulty, the items were distributed along the instrument. Although two participants mentioned that the use of gender inclusive pronouns could complicate the reading level of the items, authors decided to maintain this in compliance with the suggestions of the 7th edition of the Publication Manual of the American Psychology Association [47] and the Commission for Citizenship and Gender Equality [48] in Portugal. Lastly, three participants considered the anchors on the Likert scale 1 (Not true about us) to 7 (Very true of us) inadequate to a prospective scenario. Thus, the subscale was changed to 1 (Not probable at all) to 7 (Totally probable). The instruction to complete the scale was changed to “For each item, please, select the response you believe that will best describe the way you and your partner will work together as parents”.

### 3.2. Thematic Analysis

Thirty-four excerpts of the interviews were coded by the first author together with second and third author and with a master’s student. An inter-rater agreement of 94.12% was obtained. The coding process allowed the researchers to discuss, compare, and contrast their thematic coding, agreeing upon a final set of themes/categories [49]. At the end of this analysis, conclusions were drawn considering the relevance of the CRS-PV scale to the sample population and further considered potential item additions to the battery [27]. The thematic map of categories generated is presented in Figure 2 with each theme detailed below.

Socioeconomic Contexts. The first emergent category was the socioeconomic contexts. This category includes the economic variables that might be a barrier to achieve parenthood. The economic situation was referenced three times by two participants “For me, the major challenge is the economic situation […] and because of our economic situation, I have to work, my partner has to work, and we would need someone’s support” (Castro, gay man, 28 years old). Also, a heterosexual man raised this question: “We need somebody to help us monetarily” (Renato, heterosexual man, 26 years old).

Families of Origin. The second category, families of origin, encompassed two subcategories, namely, support, and (re)adjustment of family relations, both within the couple and with the respective families of origin. The subcategory support consisted in the valorization, anticipation, or desire to receive emotional support from both families of origin. This theme was mentioned seven times by four participants, who raised some important questions, “I can consider parenthood because my mother-in-law is available to take care of the child” (CG, heterosexual woman, 29 years old); “I have seen several persons postponing the decision of having children because they do not have enough social support” (Isabel, heterosexual woman, 38 years old); “There is a lot of things that I will always need: advice, tips, and having the family support is an added value” (Castro, gay man, 28 years old). The second subcategory, (re)adjustment of relations, was mentioned four times by two participants: “But if we had a child my relationship with my partner will need adjustment” (CG, heterosexual woman, 29 years old); “When you have a difficult relationship with your family of origin or with your partner’s family of origin, you fear to deal with them, because probably bring you with more contact with family, and not everyone is willing to do that” (Isabel, heterosexual woman, 38 years old).

Anticipation of Stigma Upon Sexual Minority Parenting. The third category, anticipation of stigma upon sexual minority parenting, includes five subcategories, namely, (i) internalization of stigma; (ii) dyadic problems; (iii) families of origin; (iv) legal structures; and (v) social challenges. The first subcategory, (i) internalization of stigma, concerns the process through which the person internalizes the stigma that they had suffered. This category was mentioned twice by one participant “As a gay couple, some situations might happen and I won’t feel so comfortable to handle them; and I can miss having a female partner in the relationship [to deflect attention]” (Castro, gay man, 28 years old); “I already had relationships with persons who had not their sexual orientation well-established neither with themselves nor with the family” (Castro, gay men, 28 years old). The second subcategory, (ii) dyadic problems, considered the influence that stigma related to sexual minority parenting could have on the couple’s relationship. Dyadic problems were mentioned four times by two participants, “The prejudice and discrimination which will pour over the couple could, obviously, influence the dyadic relationship” (Castro, gay man, 28 years old), “It could wear out a relationship. It could cause a big distress. Either people get closer together at a difficult time and try to overcome it together or it can act to wear down the relationship a lot.” (JDA, bisexual woman, 41 years old).

The third subcategory, family of origin (iii), concerned all situations where families of origin were mentioned as sources of discrimination, which was mentioned ten times across all of the sexual minority identified participants: “I have heterosexual people in my family (…) and other family members are always asking them when a child will come, and for me, even though they know that I am in a relationship (…) that question is not mentioned” (Castro, gay man, 28 years old); “A question like this [having children] countering that is difficult (…) it’s difficult to start on that point [having no family support upon parenthood], starting right there” (JDA, bisexual woman, 41 years old); “I imagine that my family would give us more support than my partner’s family and I need to be understanding about that” (Luís, gay man, 45 years old).

The fourth subcategory, (iv) structural challenges concerning disparities in services and laws on parenting in terms of parents’ sexual orientation. Structural challenges were mentioned one five occasions by two participants: “Even when starting finding health professionals, finding people who we can talk to, without being afraid to talk about this… we never know how the doctor will answer” (JDA, bisexual woman, 41 years old); “The bureaucracy (…) I have the view that this is a more difficult pathway (…) from the initial information given about the process and throughout the process itself” (Luís, gay man, 45 years old). Lastly, the fifth subcategory referred to the (v) social stigma surrounding sexual minority parents and was mentioned twice by two participants: “As if it was a bigger expectation (…) due to the fact that I am homosexual and I do not have access to biological parenting, so I have to become an exemplary father” (Luís, gay man, 45 years old); “All the comments we hear from people around, all their prejudices saying that is not the same thing (…) or that the children will learn a divergent sexuality.” (JDA, bisexual woman, 41 years old).

Scale Reconstruction. From out thematic analysis [50] above and considering our review of the relevant literature in the field, we added two subscales to the original instrument, one concerning families of origin (to be answered by all persons, independently of sexual orientation), and a second one concerning the challenges faced by sexual minority people when transitioning to parenthood. To accomplish this aim, the first author created the items, and the second and the third author, experts in the field of family and sexual orientation and gender diversity psychology, validated the items [16,45]. The subscale regarding families of origin composed of seven items, namely: (i) “It will be easier to raise our child if we have our parents’ support”; (ii) “My partner and I will disagree regarding the financial help our parents may give”; (iii) “My partner will accept the suggestions my parents give regarding our child’s care”; (iv) “We will disagree about who will take care of our child: my parents or my partner’s parents”; (v) “Our families’ beliefs regarding the raising of our child will create conflict between my partner and me”; (vi) “Our parents’ interferences in how we raise our child will negatively impact our relationship”; (vii) “Spending more time with our families of origin after becoming parents will improve the relationship between my partner and me”.

The subscale concerning the challenges faced by sexual minority persons is composed of 11 items, namely: (i) It will be difficult for my partner to speak out in public about our family”; (ii) “My partner will have difficulty coping with our child experiencing discrimination due to our sexual orientation”; (iii) “Having a child will make us more visible targets of prejudice”; (iv) “My partner will have doubts about how to be a parent because of our sexual orientation”; (v) “My partner will be less comfortable in raising our child because of not having a woman/man as a role model at home”; (vi) “It will be harder to become parents if our family does not support our sexual orientation”; (vii) After having a child, our family will support us as parents”; (viii) “Because our parents are not expecting us to have a child, they will provide us with less support”; (ix) “It will be easy to find a school which accepts all types of family”; (x) “It will be easy to find health professionals who do not discriminate LGBT families”; (xi) “It will be easy to teach our child how to deal with prejudice”.

## 4. Materials and Methods (Study 2)

### 4.1. Participants

This study encompassed 535 participants using a non-probabilistic snowball sampling in a committed relationship aged between 18 and 45 years old. We considered the limit of 45 years as a useful threshold for parenthood in the Portuguese context because (i) the age limit for access to assisted reproduction techniques (ART) funded by the Portuguese National Health Service is 42 years old, with a legally established upper age limit of 50 years old; and (ii) in Portugal, people older than 45 years seldom present themselves as candidates to adopt [51]. To conduct the analysis, the sample was divided in three groups: one group consisted entirely of sexual minority people (this group were considered in the EFA to establish the new subscales of the CRS-PV). The other two groups were composed of heterosexual people, randomly selected, using IBM SPSS version 27, from the main sample to conduct a separate EFA and also a CFA.

The sexual minority sample encompassed in total 167 participants, whose age ranged from 18 to 45 years old (M = 27.8; SD = 6.87). Regarding their sexual orientation, 40 participants identified as lesbian women, 50 as gay men, 63 as bisexual, 13 as pansexual and one as asexual. Concerning gender, 61.1% identified as women, 33.5% as men, and 5.4% as non-binary. Most participants (84.4%) lived in an urban area, 50.9% had a full-time job, and 71.9% had a university degree.

The heterosexual sample used to perform a separate EFA was composed of 198 persons whose age ranged from 18 to 45 years old (M = 26.5; SD = 5.21), and 65.6% lived in an urban area, 54.7% had a full-time job, 80.2% had university degree, 90.6% identified themselves as women, and those remaining as men. Lastly, the separate heterosexual sample (*n* = 176) used in CFA ranged from 18 to 45 (M = 26.1; SD = 5.19), and 72.2% lived in an urban area, 44.3% had a full-time job, 72.2% had university degree, and 88.6% identified themselves as cisgender women, and those remaining as cisgender men.

### 4.2. Procedure

Data were collected on-line from April to September 2020 in Portugal, as a part of a larger study about the parenthood plans and support networks of LGB and heterosexual people, entitled: “Parenthood processes and social networks among lesbian, gay, bisexual, and heterosexual persons: a dyadic, intergenerational, and cross-cultural approach”. Recruitment procedures were the same for all participants, and we acknowledge the particular contribution of LGBT organizations in disseminating the study survey. The study received the approval of the Ethics Committee of the host institution (Ref. 2019/09-08c) and was advertised on-line in websites, social media and LGBT+ organizations. The confidentiality of the collected dataset was ensured, with the survey link being accessed only via a secure university service. Completing the questionnaire took approximately than 15–20 min.

### 4.3. Measures

Participants were asked to fill in the Coparenting Relationship Scale—Prenatal Version [12] with the changes reported in Study 1. The description of the original instrument can be found in section Study 1 subsection Measures.

## 5. Results

### 5.1. EFA with the Heterosexual Sample

To assess the suitability of the sample data for EFA, the Kaiser-Meyer-Olkin (KMO) Measure of Sampling Adequacy [52], and Bartlett’s Test of Sphericity [53] were considered. Both values were acceptable and indicated a very good sampling adequacy for all factor analyses (KMO = 0.868; χ^2^ (666) = 2798.6 *p* < 0.001). Then, considering the factor structure proposed by Pinto and colleagues [12], plus the additional dimension of families of origin derived from study 1, five factors were extracted. We used Principal Component Analysis (PCA) to determine factor extraction, analyzed by the Varimax rotation. After the PCA, the model with five factors explained 47.49% of total variance with all eigenvalues higher than 1. Item 17 “We will often discuss the best way to meet our child’s needs”, presented a low communality value (<0.30) and was thus eliminated [54].

To assess the final structure of each factor, the component matrix with PCA was analyzed and items eliminated if they had either (i) lower loadings; (ii) a difference between factors lower than 0.100; (iii) or low contribution to factor [54]. Considering these criteria, five items were eliminated (items 5, 12, 15, 17, and 18).

Table 2 presents the EFA findings with five prospective coparenting dimensions and the respective loading of each chosen item. Factor 1 Coparenting Support comprised 13 items, of which 10 were present in the equivalent factor identified by Pinto and colleagues’ work [12]. Factor 2 Conflict included 10 items with half of these items contributing to the equivalent Coparenting Conflict dimension on the Pinto’s work [12]. The remaining items were part of Pinto et al.’s disagreement (*n* = 4) and lack of coparenting support subscales (*n* = 1). Factor 3 was named Conflict regarding Families of Origin and comprised four items generated in Study 1, described in the present paper. Factor 4 included two items which belonged to the subscale lack of coparenting support in Pinto et al. [12]. However, in Feinberg et al.’s work [2], these two items had been included on the subscale Closeness. Considering this, Factor 4 also was labelled Closeness. Lastly, Factor 5, Positive Influence of Families of Origin, included three items that had been generated in Study 1. Reliabilities (Cronbach’s alphas) of all subscales are presented in Table 3. Considering these values, item 7 was eliminated from the subscale Support and the low reliability of the subscale Positive Influence of Families of Origin rendered this subscale inappropriate for future use.

### 5.2. CFA with Heterosexual Sample

In a second stage of the analysis, the factorial validity of the structure yielded by the EFA was established through CFA with *n* = 176 of the heterosexual participants in study 2, using the AMOS software, version 27. Multivariate outliers were sought using the Mahalanobis squared distance (D2) (p1 and p2 < 0.001) and the highest six outliers were eliminated [55]. Six items showed high covariance values across factors and were thus discarded from further analyses (item 4 “My partner will pay a great deal of attention to our child.”; item 9 “Sometimes, one or both of us will say cruel or hurtful things to each other in front of the child.”; item 11 “My partner sometimes will make jokes or sarcastic comments about the way I will be as a parent.”; item 13 “My partner will be sensitive to our child’s feelings and needs.”; item 24 “My partner will make me feel like I am the best possible parent for our child.”; and item 30 “We will yell at each other within earshot of the child.” [56]. Lastly, the modification indices suggested covariances between some errors, namely: errors 6 and 28, and errors 25 and 3. As these items fell within the same dimensions and their content was related to the same topic, these errors were correlated [57]. Overall, in the CFA the maximum likelihood estimation, calculations indicated a good model fit (χ^2^ (201) = 281.5, *p* < 0.001; CFI = 0.940; RMSEA = 0.048; TLI = 0.931), with factor loadings ranging from 0.27 to 0.84. The final model of PCRS is represented in Figure 3. All the subscales remaining presented satisfactory composite reliability values (Support = 0.862; Conflict = 0.730; Conflict Regarding Families of Origin = 0.795; and Closeness = 0.769 [58,59].

### 5.3. EFA with Sexual Minority Sample

Concerning the sexual minority subsample, neither the Coparenting Relationship Scale [2] nor the Coparenting Relationship Scale—Prenatal Version [12] had considered the challenges associated with being a sexual minority individual. Thus, to understand the specificities of the prospective coparenting relationship challenge for sexual minority people we conducted a separate EFA [60]. Data were deemed suitable for EFA (KMO = 0.802; χ^2^ (1128) = 3695.1, *p* < 0.001). In this EFA, six factors were extracted: the five factors previously established in the heterosexual sample by EFA and CFA plus the specific dimension for sexual minority persons of anticipated stigma upon parenting generated in study 1. Principal Component Analysis (PCA) was used to determine factor extraction, analyzed by the Varimax rotation with Kaiser normalization. After conducting a PCA, the model with six factors explained 46.99% of total explained variance with all eigenvalues higher than 1 [61].

Concerning the analysis of communalities, five items were eliminated because of low communality values [54], namely: (i) “My partner will like to play with our child and then he/she will leave the dirty work to me”; (ii) My partner will not trust my abilities as a parent”; (iii) “We will often discuss the best way to meet our child’s needs”; (iv) “Having a child will make us a more visible target of prejudice”; (v) “My partner will have doubts about his/her parenting abilities because of his/her sexual orientation”.

To assess the final structure of each factor, the component matrix using PCA was analyzed using the previously established criteria [55] and five items were eliminated. Table 4 presents the EFA findings with six prospective coparenting dimensions: (i) Support; (ii) Undermining; (iii) Disagreement; (iv) Conflict with families of origin; (v) Institutional support; (vi) Support from families of origin and the respective loadings of each chosen item. The internal consistency values (Cronbach’s alphas) of all measures are presented in Table 5. Concerning the reliability analysis, item 43 was eliminated from the subscale Disagreement, and item 48 from the subscale Institutional Support, to increase their reliability. The subscale Support from Families of Origin was deemed inappropriate because of its reliability below acceptable level. Thus, the first factor, Support was comprised of 13 items all of which were included in Pinto et al.’s work. The second factor, Undermining, included four items considered in the same dimension in Pinto et al.’s study [2], and two others from the Pinto et al.’s conflict, and disagreement dimensions. In Factor 3, three items belonged originally to the disagreement dimension in Pinto et al. [12], two belonged to the conflict dimension, and the remaining item in one of the subscale generated in study 1 reported in the present paper. Factor 4, Conflict with Families of Origin, comprised the same items as in the equivalent subscale for the heterosexual sample in the present paper. However, item 42 generated in study 1 regarding the social stigma faced sexual minority individuals, also loaded on to this factor. Lastly, Institutional Support, Factor 5, included two items generated in study 1 in the present paper concerning the specific challenges sexual minority participants anticipated regarding presenting their coparenting relationship to service professionals in society.

As can be observed on Table 5, the subscale Disagreement presented a value of internal consistency of α < 0.70. This low value can be regarded as acceptable but should continue to be interpreted with caution [62].

## 6. Discussion

The aim of the present study was to refine the previous adaptation of Pinto and colleagues [12] of Feinberg et al.’s [2] Coparenting Relationship Scale to assure the adequacy and ecological validity of this instrument in assessing the perception a prospective coparenting relationship in an inclusive way to sexual minority people. In order to ensure the instruments’ ecological validity, it was important to consider the specificity of coparenting anticipation amongst sexual minority people and in societies where familistic cultural values prevail. In study 1, cognitive interviewing [43] of the instrument resulted in several changes, including: (i) linguistic adaptations to the prospective version; (ii) changing the order of some items; (iii) division partition of items portraying distinct contents (iv) rewording of some items; (v) changing the answer anchors on the Likert-type scale to prospective terms. Furthermore, after a thematic analysis of participants’ answers, the following additional changes took place (i) the inclusion of one subscale concerning families of origin to be answered by all participants; and (ii) the addition of one subscale concerning anticipated stigma challenges to sexual minority people upon parenthood. This process augmented the facial and ecological validity of the CRS-PV, which is a crucial step in the validation of an instrument [27]. Study 2 aimed at testing the factorial validity of the instrument through EFA and CFA with heterosexual participants, and an EFA with sexual minority ones. Our results indicated that prospective coparenting relations assessed through the current instrument showed different structures as a function of sexual orientation. Thus, one important finding of the present study is the need to assess future coparenting relationships using different criteria for sexual minority and heterosexual persons.

Concerning the heterosexual sample, the original subscale Undermining [2,12] was discarded in the present study. The prospective version of this subscale might have enhanced the social desirability of participants’ answers to deny the possibility of undermining as parenting is still seen as a sensitive subject. The dimension Closeness was part of Feinberg’s original scale [2] but was not included in Pinto’s work [12]. In fact, a close positive dyadic relationship may buffer parents from the negative effects of parenting stress [1,9] and coparenting support seems to strongly contribute to a consistent dyadic relationship [1,3,9]. In this regard, the heterosexual subsample of this study seemed to anticipate coparenting as a positive influence on their dyadic relationship. This assumption might not be consistent with the lower levels of dyadic agreement and well-being after having children [63]. However, the greater social pressure to become a parent experienced by heterosexuals, when compared to sexual minority people [21] might account for this unrealistic belief.

Nevertheless, in our studies there were common points among sexual minority and heterosexual participants concerning prospective coparenting. It is interesting to note that coparenting support was the most important subscale for all participants, regardless of the individual’s sexual orientation, as the support dimension accounted for the highest portion of explained variance in both analyses. In fact, coparenting support promotes parental psychological well-being as the sexual minority participants in the qualitative component of the study noted, a strong dyadic relationship might be essential to overcome the difficulties inherent to the lack of family and social support [1,3,9].

Both heterosexual and sexual minority participants in our psychometric analyses of the CRS-PV highlighted the role of families of origin. In our qualitative thematic analysis of the interviews, the values associated with familism (i.e., cooperation, intergenerational support) were often mentioned [22,24,25,29]. In the PCA analysis, it seemed that there were several similarities in this dimension as rated in both the sexual minority and heterosexual participants subsamples. Nonetheless only one item was specific to sexual minority people: “My partner will be less comfortable in raising our child because of not having a female/male role model at home”. In fact, this seems to be a type of stigma concerning sexual minority parenthood [64,65] and might be internalized [66].

Both groups of participants had considered the negative interference of families of origin on their own dyadic relationship after parenthood, probably because each generation tend to construct their own parenting behaviors differently to at least some existent compared with their family of origin [67]. This pattern could raise conflicts between generations and, consequently, disturb the dyadic relationship. However, intergenerational support has formed a central tenet of familism, and is perceived as especially important in countries where the state does not assure support needs [68]. Furthermore, parenthood as a transition in the family life cycle requires the realignment of relationships and redistribution of role functions in families. Within this realignment, couples will experience the challenge of delimitating their dyadic space [69]. In a familistic culture, this could be specifically challenging: on the one hand, couples, as parents, seek their families’ support, on the other hand, the interference of family could complicate the couples’ adjustment to parenting together.

The importance of institutional support in prospective coparenting relationship emerged only among sexual minority participants. In fact, interactions with professionals in care settings and school environments have been identified as a common source of concern for prospective sexual minority parents [23,70]. In Portugal, where the present study was conducted, higher levels of stigma are still anticipated despite the inclusive legal context concerning same-gender parenthood [71]. Regarding institutional barriers, recent studies have found that sexual minority parents successfully pursue several strategies to overcome these difficulties, such as selecting inclusive schools; being open about their families; engaging in school life; building community relationships; and improving resilience in their children to deal with difference and prejudice [72].

Surprisingly, the dimension closeness did not form a coherent factor in the analysis of scores on the CRS-PV among sexual minority people, whereas it did coalesce into a factor when we consider the scores of heterosexual participants. For some sexual minority people, just becoming a parent is difficult, this might be a postponed life project for many same-gender couples [73]. Consequently, many sexual minority people may not consider parenthood to be a key factor in the success of their dyadic relationships.

The present study is a contribution to the validation of an instrument to assess prospective coparenting relationship, considering the sexual orientation of participants, and the relationship with families of origin. In spite of its pioneering contribution, some limitations of the present study should be addressed. First, considering that this is the first study exploring the prospective coparenting relationship among both heterosexual and sexual minority persons, larger samples will be needed to confirm the structure of CRS—PV. Given the low internal consistency of the dimension Disagreement among sexual minority people [62] this factor should be treated cautiously, and future studies should address its suitability. In Study 1 lesbian women were not represented among sexual minority participants. Furthermore, in study 2, most participants were highly educated heterosexual cisgender women residing in urban areas and at the same time heterosexual cisgender men were the least represented group. These biases are commonly in research studies and should be addressed with different recruitment procedures, notwithstanding collecting a sexual minority sample has specific challenges and raise several concerns [74]. Regarding culture and its influence in parenthood [22,26] the influence of cultural values [28] in coparenting relationship should also be studied in further research in different countries. Lastly, parenting interventions in several domains [75] are needed to evaluate and train long-term great co-parenting to assure the well-being and safety of all children.

## 7. Conclusions

The results of the present study have highlighted the importance of considering the sexual orientation of participants when we are inspecting the prospective coparenting relationships. The different profile of prospective coparenting for sexual minority groups compared to their heterosexual peers in terms of the role of anticipated stigma has indicated the need to enhance resilience in the face of social prejudice against sexual minority parents. In this regard, therapists should consider the role of internalized discrimination [65] in clinical interventions with sexual minority prospective parents. Furthermore, longitudinal studies that track participants through the prospective coparenting relationship until the first years of parenting might highlight the different needs of sexual minority parents across the family life-cycle [21,74]. Other variables such as socioeconomic level [76], educational level and type of family of origin [77] or attachment style [78] that have direct implications in the beliefs associated with parenting and the exercise of co-parenting in all persons should be taking into account in future studies.

Lastly, and considering the importance of sources of support in transition to parenthood among sexual minority persons [20,21] studies inspecting the prospective plans of this population should considered the role of families of origin and the dyadic dimension of parenthood. The assessment of all these dimensions will promote the facial and ecological validity of psychometric instruments measuring the prospective coparenting relationship [27]. In Appendix A, a final version of the instrument could be consulted.

## 8. Patents

This section is not mandatory but may be added if there are patents resulting from the work reported in this manuscript.

## Figures and Tables

**Figure 1 ijerph-19-06345-f001:**
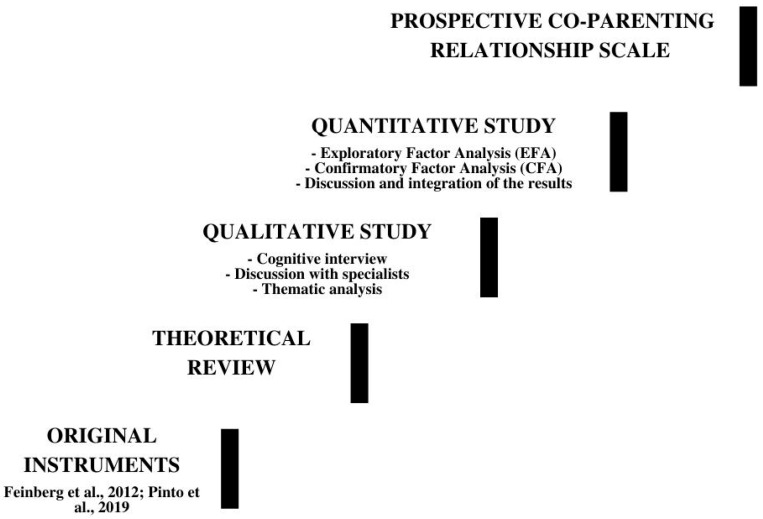
Procedure used to adapt and validate the Prospective Co-Parenting Relationship Scale [2,12].

**Figure 2 ijerph-19-06345-f002:**
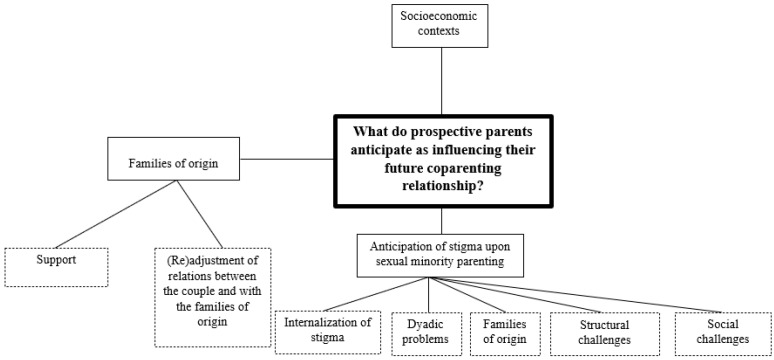
Thematic map of the themes obtained through cognitive interviews regarding prospective coparenting relationship.

**Figure 3 ijerph-19-06345-f003:**
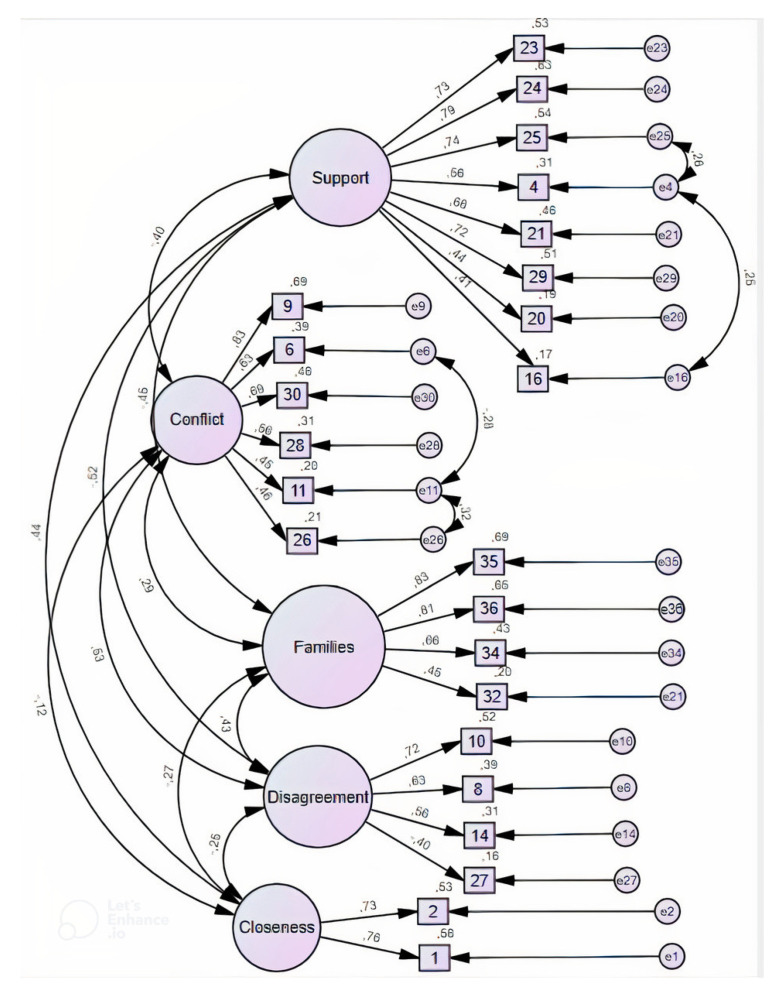
Final Factorial Structure of Prospective Coparenting Relationship Scale (PCRS) in a Heterosexual Sample, N = 176.

**Table 1 ijerph-19-06345-t001:** Sociodemographic Characteristics of Participants.

Code	Age	Residence	Gender	Sexual Orientation	Gender Identity	Educational Attainment	Marital Status	Employment Status	Ethnicity
JDA	41	Major city	Feminine	Bisexual	Woman	Master	Living together *	Part-time	Portuguese
Luís	45	Major city	Masculine	Gay	Man	Master	Living together *	Full-time	Portuguese
Castro	28	Town	Masculine	Gay	Man	Master	Living apart together **	Unemployed	Portuguese
Isabel	38	Major city	Feminine	Heterosexual	Woman	Master	In a civil union	Full-time	Portuguese
CG	29	Town	Feminine	Heterosexual	Woman	Master	Cohabitation	Full-time	Portuguese
Renato	26	Town	Masculine	Heterosexual	Man	High school	Living apart together **	Full-time	Portuguese

Notes: * In a committed relationship (cohabiting), ** In a committed relationship (not cohabiting).

**Table 2 ijerph-19-06345-t002:** Final Exploratory Factor Analysis (EFA) of Prospective Coparenting Relationship Scale (PCRS) with Varimax Rotation and Kaiser Normalization among Heterosexual Sample.

	Factors				
Item	1	2	3	4	5
(22) My partner will appreciate how hard I will work at being a good parent.	0.774				
(23) When I am at my wits end as a parent, my partner will give me extra support I will need.	0.763				
(24) My partner will make me feel like I am the best possible parent for our child.	0.713				
(21) We will grow and mature together through our experiences as parents.	0.676				
(25) I believe my partner will be a good parent.	0.650				
(4) My partner will pay a great deal of attention to our child.	0.630				
(29) My partner will tell me I am doing a good job or otherwise will let me know I am being a good parent.	0.597				
(3) My partner will ask my opinion on parenting issues.	0.581				
(19) My partner will undermine my parenting.	0.572				
(13) My partner will be sensitive to our child’s feelings and needs.	0.537				
(20) My partner will be willing to make personal sacrifices to help taking care of our child.	0.535				
(16) My partner will have a lot of patience with our child.	0.497				
(7) It will be easier and funnier to play with the child alone than with my partner.	0.473				
(8) My partner and I will have different ideas about how to raise our child.		0.699			
(10) We will have different ideas regarding our child’s eating and sleeping habits and other routines.		0.660			
(9) Sometimes, one or both of us will say cruel or hurtful things to each other in front of the child.		0.654			
(30) We will yell at each other within earshot of the child.		0.647			
(6) We will argue about our child in the child’s presence.		0.636			
(14) My partner and I will have different standards for our child’s behaviour.		0.595			
(26) Sometimes, I will find myself in a mildly tense or sarcastic interchange with my partner.		0.504			
(28) We will argue about our relationship or marital issues unrelated to our child, in the child’s presence.		0.493			
(11) My partner sometimes will make jokes or sarcastic comments about the way I will be as a parent		0.523			
(27) My partner and I will have the same goals for our child.		0.500			
(35) The way each of our families raise children will be a motive of conflict between me and my partner.			0.789		
(36) Our relationship will wear off with the interference of our parents in the way we raise our child.			0.719		
(34) We will disagree about who will take care of our child: my parents or his/her parents.			0.631		
(32) The financial help provided by our parents will be a motive of disagreement between me and my partner.			0.622		
(2) My relationship with my partner will be stronger after having a child.				0.733	
(1) Parenting will give us a focus for the future.				0.661	
(33) My partner will accept the tips that my parents will give about childcare.					0.751
(31) It will be easier to raise a child if we have our parents’ support.					0.634
(37) Spending more time with our families of origin after being parents will improve our relationship.					0.621

Notes. Factor 1 = Coparenting Support; Factor 2 = Conflict; Factor 3 = Conflict regarding Families of origin; Factor 4 = Closeness; Factor 5 = Positive Influence of Families of Origin.

**Table 3 ijerph-19-06345-t003:** Internal Consistency of Lack of Coparenting Support; Coparenting Conflict; Conflict Families of Origin; Coparenting Closeness; Positive of Influence of Families of Origin among Heterosexual Sample.

Subscale	Cronbach’s Alphas (α)
Coparenting Support	0.844
Coparenting Conflict	0.833
Conflict Families of Origin	0.717
Coparenting Closeness	0.758
Positive Influence of Families of Origin	0.499

**Table 4 ijerph-19-06345-t004:** Final Exploratory Factor Analysis (EFA) of Prospective Coparenting Relationship Scale (PCRS) with Varimax Rotation and Kaiser Normalization among Sexual Minority Sample.

	Factors					
Item	1	2	3	4	5	6
(23) When I am at my wits end as a parent, my partner will give me extra support I will need.	0.791					
(22) My partner will appreciate how hard I will work at being a good parent.	0.778					
(4) My partner will pay a great deal of attention to our child.	0.752					
(25) I believe my partner will be a good parent.	0.746					
(21) We will grow and mature together through our experiences as parents.	0.696					
(29) My partner will tell me I am doing a good job or otherwise will let me know I am being a good parent.	0.693					
(20) My partner will be willing to make personal sacrifices to help taking care of our child.	0.692					
(24) My partner will make me feel like I am the best possible parent for our child.	0.689					
(16) My partner will have a lot of patience with our child.	0.600					
(3) My partner will ask my opinion on parenting issues.	0.584					
(27) My partner and I will have the same goals for our child.	0.544					
(1) Parenting will give us a focus for the future.	0.513					
(13) My partner will be sensitive to our child’s feelings and needs.	0.511					
(18) When all three of us will be together, my partner sometimes will compete with me for our child’s attention.		0.749				
(15) My partner will try to show that she or he is better than me at caring for our child.		0.735				
(11) My partner sometimes will make jokes or sarcastic comments about the way I will be as a parent.		0.551				
(19) My partner will undermine my parenting.		0.546				
(7) It will be easier and funnier to play with the child alone than with my partner.		0.506				
(26) Sometimes, I will find myself in a mildly tense or sarcastic interchange with my partner.		0.487				
(5) My partner will like to play with our child and then he/she will leave the dirty work to me.		0.414				
(8) My partner and I will have different ideas about how to raise our child.			0.665			
(14) My partner and I will have different standards for our child’s behaviour.			0.565			
(39) My partner will have difficulties in coping with if our child is discriminated because of our sexual orientation.			0.528			
(9) Sometimes, one or both of us will say cruel or hurtful things to each other in front of the child.			0.491			
(43) It will be harder to be parents if our family does not support our sexual orientation.			0.457			
(10) We will have different ideas regarding our child’s eating and sleeping habits and other routines.			0.448			
(6) We will argue about our child in the child’s presence.			0.414			
(34) We will disagree about who will take care of our child: my parents or his/her parents.				0.806		
(36) Our relationship will wear off with the interference of our parents in the way we raise our child.				0.640		
(32) The financial help provided by our parents will be a motive of disagreement between me and my partner.				0.613		
(35) The way each of our families raise children will be a motive of conflict between me and my partner.				0.565		
(42) My partner will be less comfortable in raising our child because of not having a female/male role model at home.				0.390		
(47) It will be easy to find health professionals who do not discriminate LGBT families.					0.801	
(46) It will be easy to find a school, which accepts all family types.					0.708	
(48) It will be easy to teach our child how to deal with prejudice.					0.587	
(45) Because they are not expecting us to have a child, our parents will provide us less support in the education of our child.						0.664
(41) After having a child, our family will support us as parents.						0.616
(37) Spending more time with our families of origin after being parents will improve our relationship.						0.558
(33) My partner will accept the tips that my parents will give about childcare.						0.521
(31) It will be easier to raise a child if we have our parents’ support.						0.518

Note. Factor 1 = Coparenting Support; Factor 2 = Coparenting Undermining; Factor 3 = Coparenting Disagreement; Factor 4 = Conflict with families of origin; Factor 5 = Institutional Support; Factor 6 = Support from families of origin.

**Table 5 ijerph-19-06345-t005:** Internal Consistency of Coparenting Support; Coparenting Undermining; Conflict with Families of Origin; Institutional Support; and Support from Families of Origin among Sexual Minority Persons.

Subscale	Cronbach’s Alphas (α)	Cronbach’s Alphas (α) after Removing Item *
Coparenting Support	0.900	
Coparenting Undermining	0.745	
Coparenting Disagreement	0.657	0.687
Conflict with Families of Origin	0.709	
Institutional Support	0.714	0.795
Support from Families of Origin	0.589	

* This decision was made considering the statistical analysis of the contribution of each item.

## Data Availability

The datasets generated for this study are available on request to the corresponding author.

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
