# Peer review of "The Prospective Co-Parenting Relationship Scale (PCRS) for Sexual Minority and Heterosexual People: Preliminary Validation"

_ijerph, 2022, doi:10.3390/ijerph19106345_

Round 1
Reviewer 1 Report
This paper has an interesting topic and it seems that the authors spent a great effort to finish the study. However, the manuscript has some major problems that should be fixed. I will list some major points below:
1) I tried to understand the methodology of the study but I could not find an answer in the paper regarding the exact methodology of this paper. I mean "Is this research a qualitative research?, Is it a quantitave reasearch or is it a mixed method study?" Because it is clear that the authors collected qualitative data in study 1 and quantitative data in study 2. Thus the methodology of this research should be clarified. When this problem is solved, the authors should revise the abstract and methodology section.
2) In the qualitative part of the research, the authors stated that they chose convenient sample and 6 participants participated into the study. If I were the authors, I would have composed the sample by using simple random sampling method.
3) There are some major problems in study 2, too. For example after the EFA and CFA analysis, the authors obtained 5 factors among Heterosexual Sample. But when we examined the numbers of the items in the factors, we see that the distribution is not even. While Support factor has 8 items, the closenes factor has only two factors. It may be a problem for the validity of the subscale. The authors should justify this result by adding some extra explanations.
4) The authors repeated the EFA for the Sexual Minority Sample. But while they conducted a CFA with Heterosexual Sample, they did not conduct a CFA with Sexual Minority Sample. I could not understand why the CFA was not repeated for Sexual Minority Sample.
5) This study is based on an existing scale and the adaptation as far as it was stated by the authors (See the exact explanation: "the aim of this study was to validate an existing instrument measuring the prospective coparenting relationship with the added consideration of two previously unexplored dimensions of prospective coparenting: anticipated social support from the 63
family of origin and, in the case of sexual minority individuals, anticipated social stigma". The authors should explain what this specific research will add to the international literature in detail.
Author Response
Dear Reviewer 1,
Thank you for all your suggestions and comments. Concerning language, two authors of this paper are English-native speakers and they both performed a careful review of the manuscript.
Concerning your suggestions:
- “1) I tried to understand the methodology of the study but I could not find an answer in the paper regarding the exact methodology of this paper. I mean "Is this research a qualitative research? Is it a quantitative research or is it a mixed method study?" Because it is clear that the authors collected qualitative data in study 1 and quantitative data in study 2. Thus the methodology of this research should be clarified. When this problem is solved, the authors should revise the abstract and methodology section.” To address this important concern we had add this information in abstract and methodology section [line 14; line 132, respectively].
- “In the qualitative part of the research, the authors stated that they chose convenient sample and 6 participants participated into the study. If I were the authors, I would have composed the sample by using simple random sampling method.” For the qualitative study, the cognitive interview, i it important to have participants with similar features of the final sample and it is not expected to be representative (Willis, 2005). For quantitative study, we understand this concern and the potential of using random sampling method. However, as it is stated by Meyer and Wilson (2009) recruiting a LGBT+ sample is difficult once sexual orientation is a category of self-identification and representative samples are very complex among this population.
- “There are some major problems in study 2, too. For example after the EFA and CFA analysis, the authors obtained 5 factors among Heterosexual Sample. But when we examined the numbers of the items in the factors, we see that the distribution is not even. While Support factor has 8 items, the closenes factor has only two factors. It may be a problem for the validity of the subscale. The authors should justify this result by adding some extra explanations.” This question is quite pertinent. We need to consider that the original instruments were used with childless people. Thus, the representation of the construct of parenting and co-parenting is expected to differ with the present sample. In the case of the closeness dimension, it emerged with only two items regarding dyadic relationship. We are aware that dimensions with more than two items are desirable, however some items need to be eliminated mostly in exploratory research (Eisinga et al., 2013). Furthermore, reliability analysis of the subscale was satisfactory, suggesting it is a parsimonious and reliable subscale.
- “The authors repeated the EFA for the Sexual Minority Sample. But while they conducted a CFA with Heterosexual Sample, they did not conduct a CFA with Sexual Minority Sample. I could not understand why the CFA was not repeated for Sexual Minority Sample.” A CFA was not conducted for sexual minority sample because we did not have enough sample size to do it properly, and thus we gave priority to the performance of EFA, considering this instrument was not used in this sample before.
- “This study is based on an existing scale and the adaptation as far as it was stated by the authors (See the exact explanation: "the aim of this study was to validate an existing instrument measuring the prospective coparenting relationship with the added consideration of two previously unexplored dimensions of prospective coparenting: anticipated social support from the 63 family of origin and, in the case of sexual minority individuals, anticipated social stigma". The authors should explain what this specific research will add to the international literature in detail.” Thank you for this suggestion. We added “These additions will expand the concept of co-parenting in a more inclusive way, considering the specific challenges of LGB persons in co-parenting. Furthermore, taking into account the importance of social support in this transition of life, support from families of origin is crucial to understand the prospective co-parenting process.” (ll 65-68).
Please, see attached the manuscript with all tracked changes. We think your suggestions had improved our work.
Thank you.
Reviewer 2 Report
This is an interesting but complex study on coparenting particularly looking at a modified coparenting scale that will include minority sexual orientations. Therefore the study is attempting to make a new psychometric which is extremely difficult to validate and be reliable since the first scale seems overly simplistic a 30 item coparenting relationship scale. Nonetheless almost no medical practice including pediatrics psychotherapy internal medicine family practice psychiatry has adequate coparenting measures therefore it's important to publish a study like this what's needed is a survey of information on what it would take to get the small 30 item coparenting scale matched up to psychiatric measures and medical measures what's not addressed in this study is that parenting is almost never scientifically studied secondly it does relate to personality psychiatric status type temperament medical problems it is central to the United States in the world to make good fit help make good families so I applaud their attempt to do so but they need to mention the following issues one pediatricians need to get involved in parenting Health to all physicians need to have data on parental parental life of their patients as a social factor electronic medical record needs to give all of us a summary of many dimensions of human well-being and I doubt that parenting ever can be considered without including psychiatric type temperament cognitive and
medical health information which should be in
the caveats should include statements like this parenting is embedded into peoples personality mental health training physical health neuropsychiatric health intelligence and cognitive domains much more is needed to evaluate long-term great parenting so that our children get the best community and the best parents can offer in terms of love and well-being
Author Response
Dear Reviewer 2,
Thank you for all your suggestions and comments. Concerning language, two authors of this paper are English-native speakers and they both performed a careful review of the manuscript.
Concerning your suggestions:
“This is an interesting but complex study on coparenting particularly looking at a modified coparenting scale that will include minority sexual orientations. Therefore the study is attempting to make a new psychometric which is extremely difficult to validate and be reliable since the first scale seems overly simplistic a 30 item coparenting relationship scale.
Nonetheless almost no medical practice including pediatrics psychotherapy internal medicine family practice psychiatry has adequate coparenting measures therefore it's important to publish a study like this what's needed is a survey of information on what it would take to get the small 30 item coparenting scale matched up to psychiatric measures and medical measures what's not addressed in this study is that parenting is almost never scientifically studied secondly it does relate to personality psychiatric status type temperament medical problems it is central to the United States in the world to make good fit help make good families so I applaud their attempt to do so but they need to mention the following issues one pediatricians need to get involved in parenting Health to all physicians need to have data on parental parental life of their patients as a social factor electronic medical record needs to give all of us a summary of many dimensions of human well-being and I doubt that parenting ever can be considered without including psychiatric type temperament cognitive and medical health information which should be in the caveats should include statements like this parenting is embedded into peoples personality mental health training physical health neuropsychiatric health intelligence and cognitive domains much more is needed to evaluate long-term great parenting so that our children get the best community and the best parents can offer in terms of love and well-being.” Thank you for this important reminder. Although these topics are not the aim of our study, we understand their importance. Considering this, we added the following topic on lines 659-661: “Lastly, parenting interventions in several domains (Stewart-Brown & Schrader-Mcmillan, 2011) are needed to evaluate and train long-term successful co-parenting to assure the well-being and safety of all children.”
Please, see attached the manuscript with all tracked changes. We think your suggestions had improved our work.
Thank you.
Reviewer 3 Report
The purpose of this work is to adapt and validate a prospective version of the Coparenting Relationship Scale in a Portuguese sample of sexual minorities and heterosexual adults who did not have children and who were in a dyadic relationship.
The description of the method in the abstract is adequate. However, an improvement in the explanation of the results obtained in the study would be desirable.
In the introduction, they conceptually define co-parenting and reveal its implications for the degree of marital adjustment and parenting. In contrast to the above, they describe the negative impact that it would have on child development when co-parenting is exercised in a dysfunctional way.
On the other hand, they point out that co-parenting is a theme that has been scarcely investigated in sexual minorities. In this area, they suggest that the social support received from families of origin plays a determining role in the mental health and well-being of sexual minorities, experiencing lower levels of stress.
It is not sufficiently clear what the authors mean by the term prospective parenting experience.
The work presents a very superficial view of successful parenting, social support, parental stress and co-parenting in an Ibero-American context. An improvement in terms of its development and theoretical depth would be desirable.
Regarding the method, the selection criteria of the participants are superficial; beyond that they require non-probabilistic snowball sampling. Check and correct.
The information provided in the instruments is adequate. The data analysis strategies are relevant and consistent with the variables analyzed. Despite the foregoing, it would be advisable to better specify and synthesize the procedure used to adapt and validate the co-parenting scale (for example, describe the process through a figure).
The presentation of results is adequate.
As for the discussion, there is a process of reflective analysis around the findings obtained and their connection with the current bibliographic review. They propose the need to use different criteria to assess co-parenting based on the individual's sexual orientation. On the other hand, the positive dyadic relationship can protect parents from the negative effects of child rearing. However, higher social pressure to become a parent experienced by heterosexuals compared to people from sexual minorities.
Social support versus co-parenting was the most important subscale rated by participants, regardless of their sexual orientation. In the same way, they manage to describe the influence generated by the family of origin regarding the beliefs associated with raising children. A deeper discussion would be desirable regarding the weight of other variables (socioeconomic level, educational level, type of family of origin, attachment style, among others) that have direct implications in the beliefs associated with parenting and the exercise of co-parenting in couples heterosexuals and sexual minorities respectively.
The conclusions are adequate, highlighting the main contributions and limitations of the study.
The reference section is in accordance with editorial standards.
Author Response
Dear Reviewer 3,
Thank you for all your suggestions and comments.
Concerning your suggestions:
1) “The purpose of this work is to adapt and validate a prospective version of the Coparenting Relationship Scale in a Portuguese sample of sexual minorities and heterosexual adults who did not have children and who were in a dyadic relationship.
The description of the method in the abstract is adequate. However, an improvement in the explanation of the results obtained in the study would be desirable.
In the introduction, they conceptually define co-parenting and reveal its implications for the degree of marital adjustment and parenting. In contrast to the above, they describe the negative impact that it would have on child development when co-parenting is exercised in a dysfunctional way.
On the other hand, they point out that co-parenting is a theme that has been scarcely investigated in sexual minorities. In this area, they suggest that the social support received from families of origin plays a determining role in the mental health and well-being of sexual minorities, experiencing lower levels of stress. It is not sufficiently clear what the authors mean by the term prospective parenting experience.” To make it clearer we added “The parenting experience, including the prospective one, which comprises the planning and negotiation about parenting and the way future parents think about their future coparenting relationship [5,6], also differs according to the social and cultural context [22,24,25,26]”, lines 55-57.
“The work presents a very superficial view of successful parenting, social support, parental stress and co-parenting in an Ibero-American context. An improvement in terms of its development and theoretical depth would be desirable.” Thank you for this suggestion. In order to cover that, in spite of far from being exhaustive, we added the Schmidt et al. (2021) in lines 109-111.
“Regarding the method, the selection criteria of the participants are superficial; beyond that they require non-probabilistic snowball sampling. Check and correct.” Checked and corrected, lines 407-408.
“The information provided in the instruments is adequate. The data analysis strategies are relevant and consistent with the variables analyzed. Despite the foregoing, it would be advisable to better specify and synthesize the procedure used to adapt and validate the co-parenting scale (for example, describe the process through a figure).” Thank you for the suggestion. You can find it in figure 1.
“The presentation of results is adequate.
As for the discussion, there is a process of reflective analysis around the findings obtained and their connection with the current bibliographic review. They propose the need to use different criteria to assess co-parenting based on the individual's sexual orientation. On the other hand, the positive dyadic relationship can protect parents from the negative effects of child rearing. However, higher social pressure to become a parent experienced by heterosexuals compared to people from sexual minorities.
Social support versus co-parenting was the most important subscale rated by participants, regardless of their sexual orientation. In the same way, they manage to describe the influence generated by the family of origin regarding the beliefs associated with raising children. A deeper discussion would be desirable regarding the weight of other variables (socioeconomic level, educational level, type of family of origin, attachment style, among others) that have direct implications in the beliefs associated with parenting and the exercise of co-parenting in couples heterosexuals and sexual minorities respectively.” Thank you for the relevant suggestion! Even though it was not within the scope of this specific paper, we included some lines about the topic in the conclusion section, lines 680-683.
“The conclusions are adequate, highlighting the main contributions and limitations of the study. The reference section is in accordance with editorial standards.” Thank you.
Please, see attached the manuscript with all tracked changes. We think your suggestions had improved our work.
Thank you.
Round 2
Reviewer 1 Report
Adapting a scale or developing a new scale is a comlicated and tiring process. That requires to calculate all the posssible risks and take precautions to overcome the problems. I was convinced by your explantions and efforts to make a better version of this manuscript. I see that most of the recommendations raised by me were done by the authors. It can be accepted.